# Flood hazard potential reveals global floodplain settlement patterns

**Laura Devitt** [1] ✉, **Jeffrey Neal** [1,2,3], **Gemma Coxon** [1,2], **James Savage** [3] & **Thorsten Wagener** [2,4,5]

Flooding is one of the most common natural hazards, causing disastrous impacts worldwide. Stress-testing the global human-Earth system to understand the sensitivity of floodplains and population exposure to a range of plausible conditions is one strategy to identify where future changes to flooding or exposure might be most critical. This study presents a global analysis of the sensitivity of inundated areas and population exposure to varying flood event magnitudes globally for 1.2 million river reaches. Here we show that topography and drainage areas correlate with flood sensitivities as well as with societal behaviour. We find clear settlement patterns in which floodplains most sensitive to frequent, low magnitude events, reveal evenly distributed exposure across hazard zones, suggesting that people have adapted to this risk. In contrast, floodplains most sensitive to extreme magnitude events have a tendency for populations to be most densely settled in these rarely flooded zones, being in significant danger from potentially increasing hazard magnitudes given climate change.

Flooding causes devastating impacts worldwide, with global damages amounting to an estimated $651 billion (USD) and affecting 1.6 billion people between 2000 and 2019 alone[1]. Disastrous floods and their impacts are increasing in severity, duration, and frequency, mainly driven by population and economic growth in flood-prone areas[2,3], and by climate change[4–7]. Losses from these events could increase by a factor of 20 by the end of 21st century[8]. Thus, understanding global fluvial flood risk and identifying population exposure is crucial for impact assessment and strategic management of future hazards[9].

Climate change impact assessments regarding potential future global flood hazards commonly use a top-down, scenario propagation approach. They follow a modelling cascade starting with atmospheric projections produced by general circulation models (GCMs), which are downscaled and used to force hydrological models to produce simulations of future river flows e.g., ref. 10. These flow simulations are then used to force global flood models to produce flood hazard maps, e.g., refs. 5,6,8, which can in turn be used to derive impact estimates, such as population exposed and economic losses[11].

The use of such model cascades can be problematic as there are numerous process parameterizations and other assumptions and uncertainties at each stage of the modelling chain, creating a 'cascade of uncertainty'[12], with biases and errors inherited at each step. The precipitation outputs of GCMs are often significantly biased[13,14], particularly regarding extremes[15], which is then compounded with other uncertainties[16]. Additional modelling choices such as modelled river sizes have been shown to result in global flood exposure estimates varying by more than a factor of 2[17]. This uncertainty leads to drastically different estimates of flood hazards, with inundation extents found to vary by 80% between global flood models[18]. Each of these methodological choices fundamentally impact the portrayal of climate change impacts on future flood hazards[19,20].

Instead of aiming to derive "scenario-led" assessments of future risk, one alternative strategy is to understand where socio-hydrological systems are sensitive to change by stress testing them against a plausible range of conditions[21,22]. While this approach contains its own assumptions and uncertainties (see discussion for review

[1]School of Geographical Sciences, University of Bristol, Bristol, UK. [2]Cabot Institute, University of Bristol, Bristol, UK. [3]Fathom, Bristol, UK. [4]Department of Civil Engineering, University of Bristol, Bristol, UK. [5]Institute for Environmental Science and Geography, University of Potsdam, Potsdam, Germany. ✉e-mail: laura.devitt@bristol.ac.uk

of uncertainties), it allows policymakers to identify the change in flood hazard that floodplains are most sensitive to. When used alongside model cascades, this approach can provide a more detailed climate change impact assessment and help to target future flood risk management interventions more effectively. This approach has been used to investigate the sensitivity of river flows[23–26], but there are very few studies that assess the sensitivity of flood hazard and population exposure[27]. One recent study mapped the sensitivity of population exposure to changes in flood magnitude at the global scale[28]. However, their "downward counterfactual analysis" calculates a linear growth in exposure between event magnitudes, which is an unrealistic assumption of how societies have developed and settled on floodplains. The authors also exclude much of the global river network from the analysis as only river basins with upstream areas >5000 km² were included.

While a warming climate is expected to increase the frequency, intensity and duration of future flood hazards, human floodplain development activities are one of the main drivers for increases in damages and losses associated with flooding. Humans have a significant influence on the hydrological cycle[29], whether it is intentional or not[30–32]. Human societies co-evolve in response to floods in many ways, formally or by necessity[33] and understanding the emergent phenomena and dynamics due to this co-evolution has been the focus of many country-specific and regional scale case studies[34–38].

In this work, we provide a high-resolution stress test of the global river network to quantify and analyse the sensitivity of inundated areas (flood extent) and exposed population to plausible variability in flood magnitude. We analyse population exposure density in frequently and rarely flooded zones on floodplains and reveal previously unseen regional differences in settlement patterns with respect to flood hazards. We find that floodplains that are most sensitive to frequent, low-magnitude events have evenly distributed exposure across hazard zones, suggesting that people have adapted to this risk. Whereas for floodplains that are most sensitive to rare, extreme magnitude events, populations tend to have most densely settled in these rarely flooded zones, potentially being at significant risk from increasing hazard magnitudes induced by climate change.

## Results

We quantify whether river reaches are more sensitive to flooding from frequent, low magnitude or from rare, extreme magnitude flood events, and investigate the factors that control this sensitivity. Similarly, we also quantify the population that would be impacted by flooding across each floodplain, and thus estimate the sensitivity of population exposure to flood events of varying magnitudes. To do this, we propose a new sensitivity index which is calculated using fluvial flood hazard maps from the Fathom global flood model[39] (see 'Methods' for model description). This is the only global flood model based on a two-dimensional hydrodynamic model for floodplains that is coupled to a one-dimensional model of river channels. This level of model complexity is widely regarded as necessary for the accurate simulation of flood events as the river channel is the main conveyor of discharge and interacts with the floodplain as flood flows move both in and out of the channel[40–42]. These coupled processes are needed to simulate differences between low and high-magnitude floods, which is essential for a sensitivity analysis, such as the one performed in this study. The flood model hazard maps provide inundation extents from fluvial flooding from different exceedance probability for all rivers with an upstream area >50 km² at ~90 m resolution. We extracted the flooded extents for ~1.2 million river reaches (between 60°N and 56°S) for each exceedance probability, and then fitted a power law to the normalized growth curve (see 'Methods' for details). The exponent of this power law, $b_r$, describes the shape of the growth in flooded extent with decreasing probability and is used to describe the sensitivity of

flooding for each river reach (demonstrated in Fig. 1b). Where $b_r$ is <1, flood extents grow most rapidly during frequent, low-magnitude events. When $b_r$ is >1, flood extents grow most rapidly during rare, extreme magnitude events. When $b_r$ is -1, the flood extents grow linearly with return period.

### Sensitivity of flood hazard

Figure 1a shows the sensitivity of flood extents to increasing event magnitudes for the global river network. There is a clear spatial pattern of reaches with different sensitivities to flooding. We investigated the relationship between reach sensitivity and numerous physical and climatic variables (see Supplement Fig. S1). We found that the dominant controls on the spatial pattern of floodplain sensitivity are local topography and upstream drainage area. Figure 1c shows how reach channel slope and upstream drainage area organize the sensitivity of 1.2 million global river reaches to flooding. Channel slope and upstream area relationships have previously been used to distinguish between bedrock and alluvial reaches along river longitudinal profiles, though mostly at the local scale[43]. Global scale studies are limited but have shown strong associations between channel slope and width with upstream drainage area within river networks[44]. We have used these findings to define thresholds for channel slope to group river reaches into three floodplain types: Confined, partially confined, and laterally unconfined (see Supplement Fig. S2 for conceptual diagrams). Confined floodplains are typically found alongside steep streams in mountainous bedrock regions and represent 8% of the reaches analysed here. In these confined floodplains, flood extents typically grow most rapidly during rare, extreme magnitude events, due to the steep slopes next to the river channel constraining how flood extents can grow laterally during frequent, low-magnitude events. Partially confined floodplains are the most common type, making up 89% of the reaches analysed. They are found on transitional streams, e.g., valley bottoms, and in these floodplains, flood extents typically grow most rapidly during frequent, low-magnitude events. This is due to the relatively wide and flat terrain next to the river channel that meets a break in slope which constrains the growth of inundated area. Laterally unconfined floodplains represent only 3% of the reaches analysed. They are very wide, flat, and unbound, which allows for exponential growth in inundation area, e.g., in deltas. The extent of flooding is typically discharge-limited, making them sensitive to changes in extreme discharge, and therefore flood extents grow most rapidly during rare, extreme magnitude events.

### Global population exposure to flood hazards

To assess how physical flood hazard sensitivity interacts with population exposure, we first quantified the number of people living on floodplains (here defined as the 1000-year modelled flood extent) using WorldPop data[45] (see 'Methods' for description of dataset). We find that ~2 billion people live on floodplains globally, of which 1.4 billion live on the 100-year floodplain. The largest proportions of populations that have settled and developed on floodplains are found in North Africa, South America and South and East Asia (Fig. 2). The global distribution of reaches in each of the floodplain categories is rather even, with partially confined being the dominant type of floodplain in all regions (see pie charts in Fig. 2). However, the proportion of the population settled on these reaches differs regionally. This is critical as the impacts of changes in flood hazard will not affect the global population uniformly, instead, regions across the globe would have different experiences in how population exposure to floods would change. Populations on partially confined floodplains are most likely to be susceptible to increasing intensity of frequent flood events. They might therefore be impacted by deeper flood waters, such as those occurring in cases of overtopping of defences, leading to significant potential for damages and economic losses. In contrast,

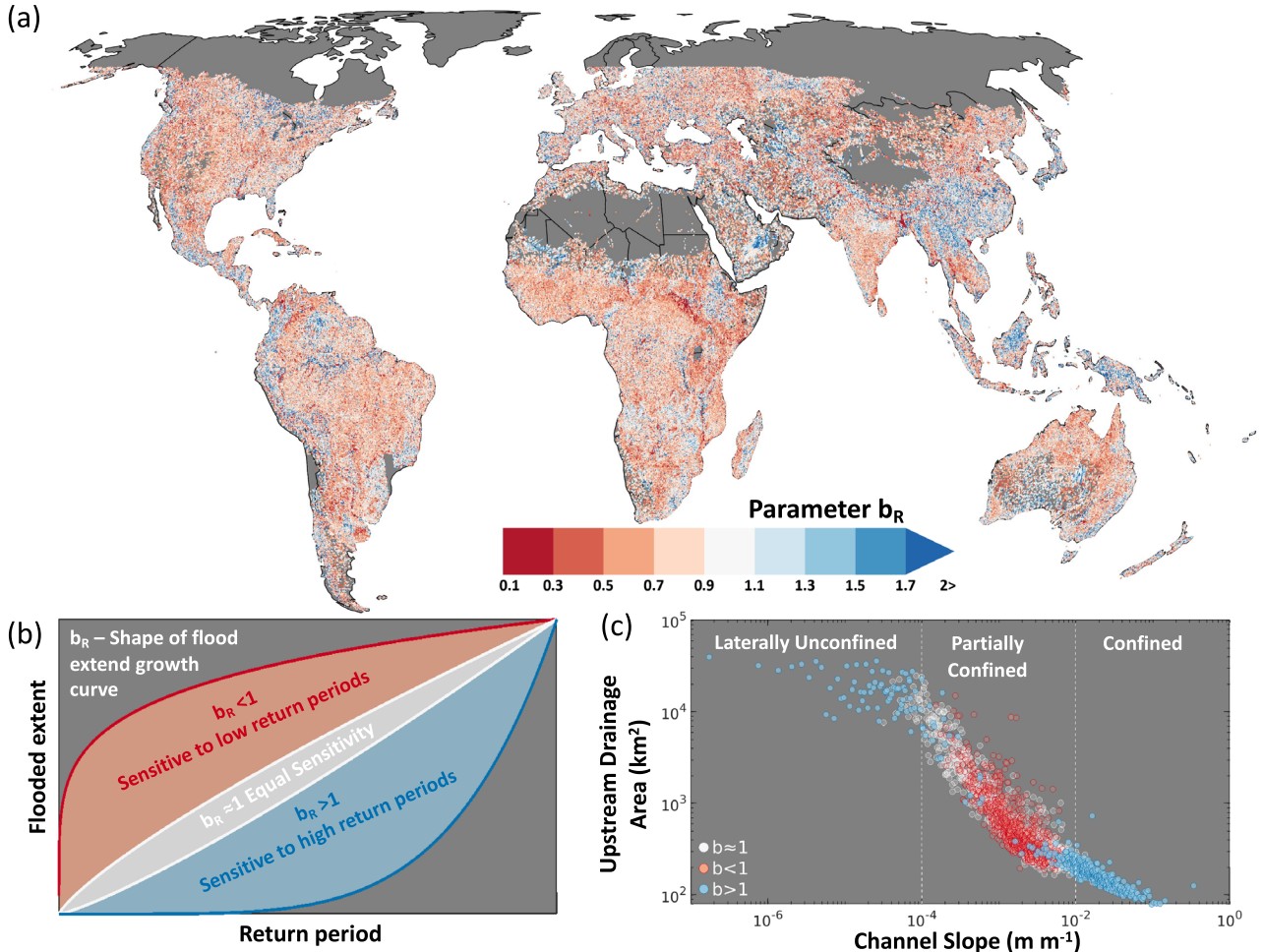

**Fig. 1 | Spatial pattern of the sensitivity of river reaches to changing flood event magnitudes.** a Sensitivity of flood extents to increasing exceedance probability for the global river network (excluding river reaches beyond 56°N). b parameters <1 indicate a sensitivity of flood extents to short return periods (100-year flood event and less). b parameters >1 indicate a sensitivity of flood extents to long return periods (100-year flood event and greater). **b** Shape of the flood extent growth curve for a range of b parameters. **c** Relationship between river reach b parameter, channel slope and upstream drainage area. River reaches have been put into 2000 bins (each representing approx. 0.2% of the data). Thresholds are placed on the channel slope to define three dominant floodplain categories: confined, partially confined, and laterally unconfined.

populations on laterally unconfined and confined reaches are more likely to be impacted by changes in the probability of extreme rare floods which would reach parts of the floodplain which were likely not inundated in living memory. Thus, providing the potential for catastrophic and unanticipated impacts.

The region with the largest number of people exposed to flooding is Asia, with 1.5 billion people living on floodplains (35% of the continent's total population). This number accounts for 75% of the world's population on floodplains, of which China and India have the largest share (490 million and 456 million living on floodplains, respectively), forming almost half of all global population exposed. Laterally unconfined floodplains such as deltas only account for 5% of the reaches across Asia. However, 20% of the population exposure is encountered on this floodplain type. These floodplains are particularly sensitive to changes in extreme river flows, and therefore a large proportion of the population could be at risk from disastrous flood events with return periods of 100-years or greater. In Bangladesh, which also has a significant number of people living on floodplains (105 million), 57% of the population exposure is on laterally unconfined floodplains, meaning many people are at risk from flooding from extreme magnitude flood events. A summary of the exposure in other regions is given in the Supplemental Information (see regional exposure on floodplains in supplement).

## Sensitivity of population exposure to flood magnitudes

To analyse the sensitivity of populations to flooding from varying event magnitudes, we also calculated a sensitivity index for exposure, $b_{pop}$, in a similar manner to $b_r$ (see 'Methods' section for full description). There are three categories of exposure sensitivity (see Fig. S3A): (1) the highest fraction of the exposed population lives in the flood zones that flood rarely, i.e. they flood only during extreme magnitude events ($b_{pop} > 1$; diagrams 1 and 2 in Fig. S3A), (2) population is approximately equally distributed across each of the flood zones ($b_{pop}$ about 1; diagrams 3 and 4 in Fig. S3A), and (3) the highest population fraction lives in the flood zones that flood frequently, i.e. they flood already during low-magnitude events ($b_{pop} < 1$; diagrams 5 and 6 in Fig. S3A). Figure 3 shows the relationship between the mean flood hazard sensitivity and population exposure sensitivity for all floodplain types at the country level. This approach allows us to visualize the interaction between the sensitivity of flood extents to varying event magnitudes, with the distribution of populations across hazard zones for different floodplain types and countries. Points along the 1:1 line represent regions where population exposure grows at a similar rate as the flood hazard for increasing event magnitude. Points on this line have the same population density in both frequently and rarely flooded zones of the floodplain. Points below this line indicate floodplains where the highest density of people is found in the frequently flooded

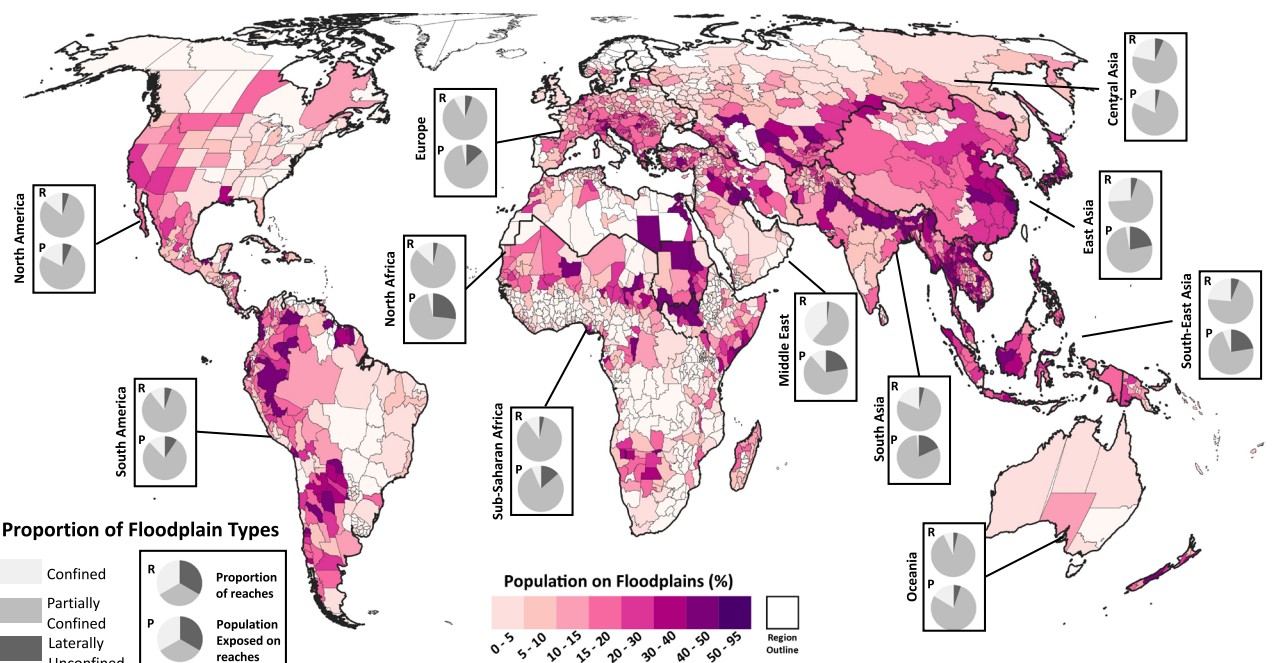

**Fig. 2 | Sub-national estimates of exposed populations.** Percentage of the population in each sub-national administrative region that live on floodplains, which is defined as the 1000-year flood extent. The proportion of the reaches that are in each of the floodplain categories for each region is shown by the top pie charts in each box. The proportion of the population settled on the different types of floodplains is shown by the bottom pie chart in each box.

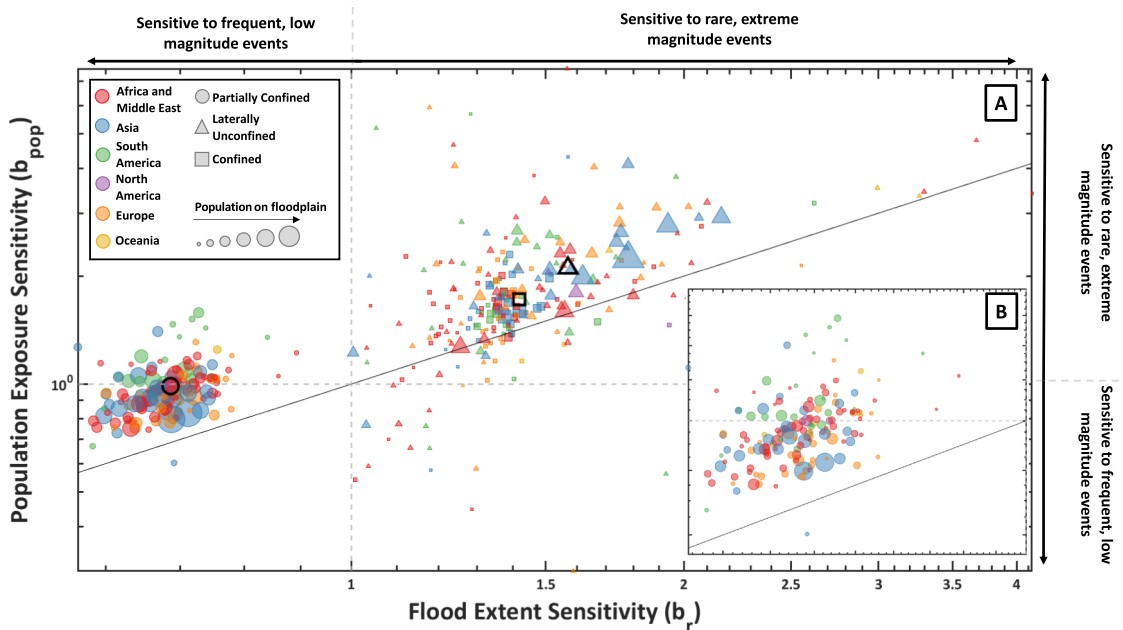

**Fig. 3 | Interaction between flood hazard and population exposure sensitivity. a** The country level mean flood hazard and population exposure sensitivity parameters ($b_r$ and $b_{pop}$) are calculated for each floodplain type. Each country has three marker shapes—one each for the mean sensitivity parameters on confined, partially confined, and laterally unconfined floodplains. The grey dashed lines indicate where the flood hazard (x axis) and population exposure (y axis) growth curves transition from growing most rapidly during frequent, low-magnitude events to during rare, extreme magnitude events. Marker closest to the 1:1 line have growth curves with similar shapes for flood extent and population, which indicates that population is distributed evenly across floodplains. Locations above the 1:1 line have greater population densities on rarely inundated areas of the natural floodplain, while locations below the line have greater population densities in frequently flooded areas. Where $b_{pop} < 1$ more people, in absolute terms, live on the frequently flooded floodplain, while $b_{pop} > 1$ indicates more people live in rarely flooded areas (see examples (4) and (3) in Fig. S3B). Point shapes represent the floodplain category for each country, the sizes have been scaled based on the number of people living on the different floodplain types in each country, and the colour refers to the region/continent that the country is in. The bold open shapes represent the centroid of all the data points for each of the floodplain types. **b** Close view of the partially confined floodplain results. Note: Continent specific plots are included in Fig. S5 for ease of visualisation.

zones of the floodplain. Locations represented by points above the 1:1 line indicate floodplains where the highest density of exposed population are found within the rarely flooded areas.

For partially confined floodplains, all countries, except for one outlier (East Timor), are located above the 1:1 line, indicating that population density is greater on rarely flooded zones of the floodplains. However, there are many countries that have a $b_{pop}$ below 1, mostly in Europe and Asia. This means that in absolute terms more people live on frequently flooded zones of floodplains because they cover a greater area. This indicates that people have either found ways to cope with flooding from frequent, low-magnitude events, or have taken on risk through choice or necessity[46]. In Fig. S4 (supplement), countries have been sized based on their mean standard of protection extracted from the FLOPROS global flood defense database[47], which reveals that some of these countries that have larger populations on frequently flooded floodplains and also have high standards of protection. In Europe, 22 countries (58%) have a mean standard of protection of at least a 50-year return period, while 10 countries (26%) have very high standards of protection of at least a 100-year return period (Austria, Belgium, Germany, United Kingdom, Croatia, Hungary, Ireland, Netherlands, Poland, Portugal, Romania, and Slovakia). 39 million people are settled on partially confined floodplains in these countries, accounting for 48% of the total exposure on this floodplain type in Europe. It is likely that in these countries significant investments have been made to mitigate the impacts of flooding from frequent flooding. In Asia, the FLOPROS database suggests that the standard of protection in many countries is low, even though the number of people exposed in frequently flooded zones of partially confined floodplains is high (1.1 billion people living on this floodplain type). This could be due to population pressure in rapidly expanding urban areas[48,49] or due to a reliance on frequently flooded floodplains for agricultural practices[50]. However, the FLOPROS database has very little data in these regions and might underestimate the degree of adaptation present.

Many countries in South and Central America exhibit different settlement patterns, where the highest density of population exposure is found in the rarely flooded zones of the floodplain. These countries include Brazil, Haiti, Paraguay, Costa Rica, Suriname, Puerto Rico, Argentina, and El Salvador. In these countries, the mean $b_{pop}$ is >1, which means that while the flood extents are growing most rapidly during frequent, low-magnitude events, the population exposed is found most densely in rarely flooded zones to the extent that the absolute numbers exposed are greater there (see example (1) in Fig. S3B). This indicates that people are more likely to mitigate risk posed by frequent flooding by living away from rivers in these parts of the world (see Figs. S6 and S7 for analysis of floodplain population density).

For the confined and laterally unconfined floodplains, most countries are also located above the 1:1 line. There are examples where the population is most densely populated in the frequently flooded zones, for example, the Dominican Republic, Zimbabwe, Cuba, Panama, Togo, Syria, and Greece. However, these places have very small population totals in these floodplain categories (<10,000 exposed) and they only make up a very small proportion of the total exposure in the country (all <1%). On steep confined reaches, many countries are close to the 1:1 line, indicating an even distribution of population across flood hazard zones. For laterally unconfined floodplains (deltas), Asian countries have the largest total populations exposed (~312 million). These people are found above the 1:1 line meaning the highest density of exposure is found in the rarely flooded zones (see example (2) in Fig. S3B for example growth curves). These results imply that people are preferentially settling and developing on parts of the floodplain that they deem to be at low risk. China, India, Bangladesh and Vietnam have the largest populations settled on laterally unconfined floodplains, with over 160 million people settled in

the rarely flooded zones of the floodplain (e.g., >100-year return period). There are few countries whose points plot below the 1:1 line but which have $b_{pop}$ values >1 (i.e., where the population density is slightly higher in the frequent flood zones—example (3) in Fig. S3B). This indicates that, in terms of absolute population numbers, the development on frequently flooded areas of laterally unconfined floodplains is less prevalent than on partially confined floodplains.

## Discussion

In this study, we have quantified the sensitivity of flood hazard and population exposure at the global scale. Our results contribute to the understanding of global flood risk, and the relationship between rivers, their floodplains, and human settlements. We find clear regional differences in settlement patterns that are related to whether floodplains are most sensitive to flooding from frequent and low magnitude, or to rare and extreme magnitude events.

Flood extents on partially confined floodplains grow most rapidly during frequent, low-magnitude events. On this floodplain type, nearly all countries are found to have the highest density of exposure in the rarely flooded zones of the floodplain. However, we find regional differences in whether more people are living in the frequently or rarely flooded zones. In Europe, countries exhibit a mostly even distribution of population density throughout all flood zones, with population totals being greatest on areas of the floodplain that would naturally flood frequently given that they make up the largest areas. For Europe and North America, this is likely due to significant investments in structural defences to protect against flooding from frequent events (see analysis of global protection standards in Fig. S4). However, structural protection tends to encourage development in flood-prone zones, a socio-hydrological process known as the "levee effect"[51,52]. This effect can lead to floodplain settlements becoming vulnerable to low-probability but potentially high-consequence flood events[51,53–55] which is problematic under climate change induced non-stationarity of flood extremes. This effect was for example seen during the catastrophic 2005 flooding of New Orleans brought by Hurricane Katrina[56] and the devasting impacts of the European floods of 2021[57].

Asian countries have a similar settlement pattern on partially confined floodplains. Given the particularly high density of population on this floodplain type in this region—14% of total land area experiences flooding but 35% of the total population are settled here (see Fig. S7)—it is likely that pressures on land have resulted in people settling in areas of the floodplain at risk from frequent flooding. A study that analysed urban growth between 1985 and 2015 finds that growth of settlements in high hazard areas has been most rapid in Asia. In fact, 'no risk' settlements expanded by about 100%, while 'very high risk' settlements expanded by over 160%[49]. As the economies of these countries grow rapidly and safe locations in urban areas become increasingly scare, new developments are built on hazardous and often cheap land to match the pace of population growth[58,59]. Another study that analysed the prioritizing of floodplains for development and farming finds the percent of cropland area in floodplains in East, Southeast and South Asia to be 51%, 67% and 59%, respectively[48]. This large fraction highlights the reliance on frequently flooded land for agriculture in the region and the necessity for human presence in floodplains. Although the FLOPROS global flood defence database suggests a far lower standard of protection (see Fig. S4) in this region compared to Europe and North America, the database is sparsely populated, and we have low confidence in the information provided. Given the very substantial exposure to frequent flooding in Asia addressing this data gap should be a priority.

South America is the only continent where more people live in rarely flooded areas of partially confined floodplains relative to the frequently flooded areas. These countries also have the highest

population density in these low hazard zones. Standards of protection are generally low, however, the relatively low population density on floodplains in general (see Figs. S6 and S7) has allowed for preferential settlement in less hazardous locations. In Fig. S7 we find around 24% of total land area in South America has potential for flooding, but only 13% of the total population lives there. Further, there is empirical evidence that during flood-rich periods impacts of flood events are reduced when an event of similar magnitude had occurred not long before. An example of this are the floods that occurred during the 1982–83 and 1991–92 ENSO events on the Paraná River. Flood damages and economic losses were reduced by 80% in the later event[60]. This dynamic has been termed the "adaptation effect"[38], which refers to the increased coping capabilities of a population due to their experiences of earlier flooding, unless subsequent flooding is more extreme[61]. The combination of relatively low population pressure on floodplains and regular experience of flood-rich periods might have facilitated relatively hazard averse settlement patterns. Our results contrast with the findings of a recent analysis of the sensitivity of flood exposure[28], where populations in South America are found to be most sensitive to flooding from frequent flood events. This difference is likely due to the choice of global flood model and methodological choices. The previous study used the JRC flood model[62], which has a coarser spatial resolution (~1 km) and only includes rivers with an upstream area >5000 km². The reaches that are included are known to show a significant overprediction bias due to the models' resolution and a lack sensitivity to event magnitude (about 5 times less than Fathom Global Flood model which is used here)[18,63]. To mitigate for overprediction of frequent flooding, flood exposure was set to zero below the standard of protection return period extracted from the FLOPROS database, essentially identifying floodplain settlement outside of Europe and North America as relatively sensitive to frequent flooding.

Laterally unconfined floodplains experience flood extents that grow most rapidly during rare, extreme magnitude events. We find that there is a strong tendency for populations to have preferentially settled on these rarely flooded zones worldwide. Although only making up 5% of the populated river reaches in our analysis, laterally unconfined floodplains are home to ~412 million people, thus accounting for 21% of the total population exposure globally. The highest population density on laterally unconfined floodplains is found in Asia and North Africa. Figure S8 (supplement) shows that these regions also have the highest density of exposure per river reach on these floodplains, with 4519, 2536, 1973 and 902 people exposed per kilometre of reach length in South-, East, South-East Asia, and North Africa, respectively. This is for example compared with only 183 and 247 people exposed per kilometre of reach length in North America and Europe. With potential increases in the frequency of extreme river flows in the projected future for these regions[64], an exponential growth in population impacted by flooding could be experienced in locations that typically would not have flooded within living memory. For example, in China, 82% of the reaches with laterally unconfined floodplains have the highest density of exposure found in the rare, extreme magnitude flood zones (69 million people living in the 100-year flood zone or greater). These regions have experienced rapid population growth, urbanisation, and industrialisation in the last two decades, including the growth of some of the world's largest urban agglomerates on large river floodplains and deltas, such as Shanghai, Cairo, Dhaka, and Bangkok. Urban population is expected to continue to grow in the future with 90% of the projected increase taking place in Asia and Africa[65]. This continued growth will put additional strain on the existing floodplain settlements, placing even more people at risk from flooding in the future.

These results provide new insights into the sensitivity of floodplains and their populations to the potential of changing flood hazard.

They are helpful to identify the greatest potential for significantly larger flood damage if flood magnitudes should increase anywhere across the world. These results therefore provide a first-order global guide for policy making on how a change in hazard might interact with exposure. Adaptation will be a critical factor in determining the severity of impacts from flood hazards in the coming decades[27]. The approach demonstrated here can provide crucial information for global agencies on where it would be most beneficial to implement flood management strategies—guidance that is urgently needed for a wide range of hazards[66]. Recent devasting floods across the globe have sharply brought into focus the urgent need to make societies more resilient to flooding now and even more so in the future[1]. Effective flood risk adaptation efforts must be based on a robust understanding of the physical dangers posed by the hazard and the increasing exposure to these events. In most regions, partially confined floodplains reveal a distribution of population across flood hazards zones that means more people live on floodplain areas that would naturally flood frequently. These populations are most likely to be susceptible to increasing intensity of frequent flood events and deeper flood waters, which may lead to significant damages danger and risk to human life if existing structural protection is insufficient. This was shown to be a major problem during the European floods of 2021 where river flows exceeded the 400-year magnitude in some places and resulted in over 200 fatalities and $46 billion in damages[57]. Such floodplains would benefit from restricting development on areas of the floodplain and from investments to raise the standard of protection of existing defences. Other measures, such as enacting zoning regulations and enhancing building codes may contribute to reducing flood damages[67].

Populations on laterally unconfined floodplains are typically most densely settled in the rarely flooded zones of the floodplain (i.e., greater than the 100-year flood event), with many people living in areas that may not have experienced large-scale flood events in living memory. 76% of all populations settled on laterally unconfined floodplains are found in Asia (~312 million people), which is particularly prone to flood hazards. Recent disastrous flooding, such as the 2017 1-in-200-year flood event in South Asia, impacted over 40 million people and killed over 1000[68], highlighting the need for investment in adaptation strategies to extreme magnitude events in this region. One of the key challenges is how to address the role of individual perceptions of risk and how these perceptions influence risk-reducing behaviour[67]. Bangladesh and Vietnam are typical examples of societies 'living with floods'[50,69] with people adapting to regular flooding by adjusting economic activities to benefit from regular inundation, e.g., through farming and fisheries[38,50,70].

Our study has been motivated by a need for complementary methods to assess global scale flood risk that are independent of at least some of the uncertainties associated with the traditional, top-down model cascade approach[27]. We therefore chose an exploratory modelling framework in which we stress test the socio-hydrologic system of global floodplains. However, our methodology and subsequent analysis are of course not free from their own uncertainties and assumptions. We have used simulated fluvial flood hazard maps from a state-of-the-art global flood model to calculate our sensitivity index, which contains its own uncertainties, originating from the quality of the elevation dataset used, knowledge of river morphology and choice of hydrodynamic model[71]. How discharge grows with return period will influence the shape of the flooded area and population sensitivity curve. Thus, one notable source of uncertainty is the use of regional flood frequency analyses[72] to estimate the growth in discharge towards more extreme conditions from pooled gauged discharge data in the Fathom flood model. These methods are limited by their need for gauged discharge data, themselves most uncertain during measurements of out-of-bank flows[73,74], and are limited by the length of observation periods, meaning statistical

extrapolation methods are needed to estimate the probability of as yet unobserved extremes[75,76]. Discharge observations are particularly sparse in arid areas, and analysis of extreme flows used in our model, and other flood models, have shown that uncertainty in extreme river flows is greatest here[16]. Furthermore, validation of the global flood model used here demonstrated that the accuracy of estimated flood hazard is lowest in arid areas relative to other climate zones[77]. The alternative to gauge-based methods is using discharge simulated by global hydrological models, however, these flows are typically generated with a lack of data to calibrate and parameterise models. They are also driven with meteorological data that poorly represent extremes[78–80], leading to predictions that are highly variable between models[16].

One key assumption is that the shape of the discharge growth curve will remain the same in the future[80]. Some studies have found differing trends in the median flood and more extreme return periods (i.e., 100-year flood) due to changing flood-generating processes in some regions[27,81–83]. However, an extensive review of flood estimation guidance in different countries[84] reveals a strong focus on using multiplicative scaling (uplifts) applied directly to peak flow rates uniformly across return period[85]. This assumption is often made due to a lack of data and poorly understood impacts of future climate change on flood-generating processes. Hydrologic non-stationarity has been discussed[86–88], but incorporating the multiple sources of changes to flood hazard (e.g., land-use changes and climate change impacts on flood-generating processes) into specific flood estimation methodologies and carrying this into meaningful guidance to be used by decision-makers remains a major challenge[89–93]. We assume that the shape of the growth curve between flood extend, or population exposed, and return period is representative for current as well as for potential future conditions (in line with current hydrological practice as discussed above). Under this assumption, our sensitivity index remains a representative indicator which allows us to understand whether a floodplain would be impacted more by an increasing probability of extreme events or a change in intensity of frequent flood events.

Another issue faced by global flood hazard models is the challenge of accounting for flood defences, especially in regions with high protection standards. Although we consider the impact of flood defences in this study, our use of undefended fluvial flood maps means we define sensitivity before these are considered. This will have overestimated exposure in countries with extensive flood protection. However, existing flood defence databases, e.g., FLOPROS[47] are, at best, informative at the national scale, as they do not make distinctions for different floodplain types and can only provide data to be superimposed onto global flood hazard maps[94]. Improving these databases to include data on defences in places where we currently have little knowledge, e.g., Asia, is crucial as this is likely to be the greatest uncertainty in global flood exposure estimates. Also, expanding these datasets to include data on spatial planning and flood zoning policies implemented as adaptation measures would be incredibly beneficial for continuing to develop our understanding of floodplain settlement patterns.

The results presented here are based on the sensitivity of river reaches and population exposed to fluvial flood hazards. We have not included other types of flood hazards, such as floods from storm surges or coastal flooding. This will be an important factor in many coastal river reaches, deltas and in regions that experience compound flood events, i.e., where fluvial and coastal flooding occurs simultaneously[95,96]. A recent study of flood risk and its interaction with poverty[97] included fluvial, coastal and pluvial flood hazard types in its estimate of global flood exposure. The authors found 1.8 billion people exposed to 1-in-100-year floods, compared with 1.4 billion exposed to fluvial flooding. Our results have only focused on population exposure to flooding. Future work could use the proposed methodology to

calculate the sensitivity of other assets, for example infrastructure, traffic systems, or public and private property[11,63,98,99]. While the results shown here represent a large proportion of the global river network, they are also not complete. We have only included rivers with an upstream drainage area >50 km², as this is the threshold for inclusion of rivers in the global flood hazard model used. We quantified the sensitivity of ~1.4 million river reaches and removed 12% of reaches due to a poor power law fit. We deem these reaches to have relationships between flood extent and event magnitudes that are more complex, and therefore additional work is needed to look at their behaviour and sensitivity. However, the reaches included in our analysis cover 89% of the global population living on floodplains.

## Methods

### Global flood hazard data
To investigate the sensitivity of global inundation extents to changes in flood magnitude, flood hazard data from the Fathom global flood model are used. Flood hazard for ten return periods (5-, 10-, 20-, 50-, 75-, 100-, 200-, 250-, 500- and 1000-year) with a horizontal resolution of 3 arc s ~90 m was simulated using the global flood model methodology of Sampson et al.[39]. The global flood modelling framework begins with extreme river flows being estimated using a regional flood frequency analysis (RFFA)[72] applied to GRDC gauging station data. These fluvial model boundary conditions are routed through 1D subgrid-scale river channels based on MERIT Hydro[100,101] and a river bathymetry estimation routine[42]. To simulate floodplain inundation, the 1D model is coupled to a 2D hydrodynamic model for out-of-bank flows. All hydrodynamic calculations are based on the LISFLOOD-FP model, which solves a local inertial formulation of the shallow water equations[101–103]. Elevation data used is from the MERIT DEM (~90 m resolution)[104] that corrects for multiple errors, including absolute bias, stripe noise, speckle noise, and tree and building height biases. Inundation simulations are made for all rivers with a drainage area >50 km². We have used the undefended flood maps, which do not include flood protection structures. No new model components are introduced in this study beyond those that have previously been described and validated[39,105–107].

### Global population data
The data used here is the 2020 WorldPop constrained population counts dataset[108]. WorldPop uses a complex model to disaggregate population over an area[29]. It uses a random forest model and a number of ancillary datasets to dynamically weight the distribution of census data over a ~90 m gridded area[108]. We processed the country-level GeoTiff data provided by WorldPop to match the 10° × 10° tile format of the global flood hazard maps.

### Quantifying sensitivity of flood extents to flow magnitude
To quantify the sensitivity of flooded extent to increasing flow magnitudes a generalised methodology has been developed. The global flood model produces hazard maps of flood depth at ~90 m resolution for ten return periods. Each grid was firstly converted to binary flood extents using flood depth threshold of >0 m. Inflow points are defined along the global river network as the location for the boundary conditions of the hydraulic modelling component. These were extracted and assigned as upstream and downstream points, and this was used to define each reach along the river network. A flooded area was calculated between each upstream and downstream segment of the river network for each of the return periods (see Fig. S8 for a diagram). The reach length is used as an effective search radius along a diagonal transect between the upstream and downstream points. For each inflow point, the flooded area was derived for each return period and normalised. A power law with the form: $F = RP^{b_r}$, where $F$ is the normalised flooded area, $RP$ is the log normalised return period of the flood event and $b_r$ is the exponent, was fitted to the data. We chose to fit the curve between the return period and flooded extent to produce

comparable sensitivity parameters at all sites by removing the influence of river flow magnitude.

The '$b_r$' parameter of the power law is used to describe the shape of the flood extent growth curve and is used as the measure of floodplain sensitivity to increasing return period. When $b_r$ is <1, the growth curve has a concave shape and grows most rapidly during the low return periods. When $b_r$ is >1, the growth curve has a convex shape and grows most rapidly during the high return periods. When $b_r$ is 1, the growth curve is linear. An $R^2$ value was calculated to measure the goodness-of-fit of the power law. Sites with an $R^2 < 0.9$ were removed from the analysis. Of 1,380,430 inflow sites across the domain, 161,947 (12% of sites) were excluded.

### Quantifying the sensitivity of population exposure to flow magnitude

Human exposure here is defined as the intersection of the flood hazard data with a spatially distributed population dataset. We multiplied the binary flood extent layers with the WorldPop data to obtain the number of people exposed for each return period. Using the same method as for the flood extents, we extracted the total number of people exposed on each river reach. We then calculated the sensitivity of the population exposure by first normalising the totals and fitting a power law of the form: $E = RP^b{}_{pop}$, where $E$ is the normalised exposure, $RP$ is the log normalised return period of the flood event and $b_{pop}$ is the exponent.

The '$b_{pop}$' parameter of the power law is used to describe how population exposure grows with increasing return period of the flood hazard. When $b_{pop}$ is <1, the growth curve has a concave shape and grows most rapidly during the low return periods. When $b_{pop}$ is >1, the growth curve has a convex shape and grows most rapidly during the high return periods. When $b_r$ is 1, the growth curve is linear.

### Data availability

Fathom global flood model data are available for academic purposes and were provided by Fathom. The WorldPop unconstrained high-resolution population counts are available to download online (https://hub.worldpop.org/geodata/listing?id=29). The FLOPROS database is available to download from the supplementary material of the dataset description paper (https://nhess.copernicus.org/articles/16/1049/2016/).

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

## Acknowledgements

This work is funded as part of the Water Informatics Science and Engineering Centre for Doctoral Training (WISE CDT) under a grant from the Engineering and Physical Science Research Council (EPSRC), grant EP/L016214/1. J.N. was supported by NERC Grants NE/S003061/1 and NE/S006079/1. G.C. was supported by a UKRI Future Leaders Fellowship Award [MR/V022857/1]. Support to T.W. was provided by the Alexander von Humboldt Foundation in the framework of the Alexander von Humboldt Professorship endowed by the German Federal Ministry of Education and Research.

## Author contributions

J.S. processed the global flood model data and provided the flood hazard data and inflow boundary conditions. L.D. conducted the sensitivity analysis and produced the figures, with input from J.N., G.C. and T.W. L.D. prepared the manuscript with contributions from all authors.

## Competing interests

The authors declare no competing interests.
