## [Peer Review File · Nature Communications]

Flood hazard potential reveals global floodplain settlement patternsREVIEWER COMMENTS

Reviewer #1 (Remarks to the Author):

REVIEW: Flood hazard potential reveals global floodplain settlement patterns

The paper addresses the important topic of how (future-) population exposure influences flood risk. This is usually done with a cascade of models, which face a challenge due to uncertainties in parametrizations. The study suggests to use an alternative approach to scenario based studies, and instead apply a stress-testing method under a plausible range of conditions.

The study focuses on stress testing the global river network to quantify and analyse the sensitivity of inundated areas (flood extent) and exposed population to plausible variability in flood magnitude. The analysis reveals regional differences in settlement patterns with respect to flood hazards. It adds to existing studies as to assess the sensitivity of flood hazard and population exposure.

The method used is based on the well-known Fathom model, and an index B showing the inundation extent for all rivers, for each RP: With a power function growth of the flood extent is described with decreasing probability –this is then assumed as an index for sensitivity of flooding. A low value shows areas where flood extent grows rapidly during frequent (low magnitude) events, and vice versa

Results show that the dominant control on the spatial pattern of floodplain sensitivity is local topography and upstream drainage area. Results show also a relationship between sensitivity and slope steepness –where delta's see the most rapid growth of flood extent during rare events.

The 1000-yr flood extents are then overlaid with worldpop data. The study finds that approximately 2 billion people live on floodplains globally, of which 1.4 billion live on the 100-year floodplain. The study makes a case showing that the type of basins varies regionally (confined, partially confined, etc), and thus the sensitivity also varies regionally. However, much of the population in e.g. Asia lives in unconfined flood plains (delta's) and the research shows these communities are at risk from extreme magnitude floods.

\ Although it is interesting to see how the authors have classified river reaches, thus far the paper does not show any new insights: steep sections of river have less flood extents and higher frequencies; delta's have larger flood extents related to low probability events.

The interesting part lies when exposure is introduced and the discussion about risk aversion and protection levels. This is done by introducing a parameter that shows how exposed people are distributed across the three flood zones. Figure 3 shows the relation between exposure distribution and flood sensitivity. The 1:1 line represents where population exposure grows at the same rate as the flood hazard with increasing event magnitude

 There is maybe a fundamental issue here regarding the method in Fig 3 (which is the central figure of the paper): isn't it too obvious that if a region has a lot of unconfined river reaches, you also may expect a larger proportion of your population living in flood zones that are un-confined? It seems like you have established a graph that double counts the flood zone type? So to then simply conclude that those people are risk averse is perhaps too straightforward: maybe they didn't have any choice. As an example: people in the overall unconfined flood zones of Bangladesh can-not all move to India or to the Chittagong hills: are they then risk averse?

Other remarks:

It took me a while, combining Fig 3 + Fs3, and I believe it shows that people have taken flood protection measures (or spatial planning forced them to live in safer areas with less flood depth?) because they generally are above the 1:1 line

 Can you perhaps address whether this latter statement is correct, and elaborate a bit? Here you could say something about whether spatial planning policies and flood zoning have had prohibiting people from building in dangerous areas?

 In the caption of S3: maybe add that in the left part, the high frequency events dominate, hence the wider 1/10 colours: correct?

Regions above the 1:1 line indicate risk averse populations.

 I am not sure whether you can make such statement or what you mean by this? Because these location are above the 1:1 line, they have invested more in flood protection, and therefore they are more risk adverse?

Lines 154-157: these are a bit unclear to me, but it also can be the case I have misunderstood the figure 3:

"Markers below this line indicate a relatively risk-taking population, where the highest density of the population exposed is in the frequently flooded zones of the floodplain – suggesting that society has taken appropriate steps to protect its population from the flood hazard through levees or other means."

 I'd say: the risk takers are indeed the ones who live in the high frequency/low impact zones (this is debatable as well by the way), but then the sentence in addition doesn't fit here ("suggesting society ... other means"). The risk takers usually not invest in flood protection?

Lines 167: inclusion of FLOPROS

 ...countries that have this patterns..": Which pattern?

 Caption S4: are you suggesting that countries with a higher protection standard are above the 1:1 line?

Line 363: did you run a sensitivity analysis for this power-function: I'd like to see how this plays out for the position of the 1:1 line?

- I am not sure what the authors mean by the sentence in lines 43-45. I expect a response to the previous paragraph, where the cascading modelling approach is challenged due to modelling biases. But then lines 43-45 speak about the need to "....identify exposure of populations and assets across a spectrum of plausible events to derive actionable information under large uncertainty".

 This is also possible with a modelling approach. I'd even argue that with a scenario approach, you may assess an even larger range of uncertainty

- Lines 124-127: "Populations on partially confined floodplains are most likely to experience vulnerability to increasing intensity and magnitude of frequent flood events, whereas confined and laterally unconfined floodplains are more vulnerable to the increasing probability of extreme flood events"

 I don't see the link directly? Doesn't the vulnerability it also depend on how much flood protection has been installed?

 I would make a difference between sensitivity and vulnerability: these are two different concepts.

- Why didn't you include flood depth? Now you see in Figure 2 that a large share of the people in SE Libya live on flood plains: but are these areas in the middle of the Sahara really hazardous flood plains?

Reviewer #2 (Remarks to the Author):

The paper by Devitt et al. analysis global patterns of human settlements in flood-prone areas. They found population to be more densely settled in areas prone to rare (but potentially catastrophic) floods, and more evenly distributed exposure in areas exposed to frequent (mostly low magnitude) floods. This work is technically sound and the paper is nicely written. I found these results not only scientifically relevant, but also socially important. They have the potential to substantially contribute to advance our fundamental knowledge on flood risk dynamics. Here, I provide three main suggestions to further enrich the description of this global study.

1. The outcomes of this paper are strongly related to recent findings on human-flood interactions.

For example, the AGU journal Water Resources Research published "Debates –Perspective on Socio-Hydrology" in 2015 to stimulate discussion on frontier research. These debates include an original research article, four commentaries, and an editorial. The former provides case studies and model results of two dynamics, i.e. adaptation and levee effect, which strongly relates to the main outcomes of this paper. I think this paper would highly benefit from including a paragraph in the introductory part describing these general dynamics of human-water systems.

2. To give perspective to these outcomes, I think it important to discuss their implications for the ongoing development of theories of human-flood interactions. As stated in the previous point, these global patterns nicely complement numerous case studies and system dynamic models carried out by the several scholars, such as the ones engaged with the international initiative Panta-Rhei-Everything Flows (IAHS). I suggest to add one paragraph and discuss more explicitly similarities and differences between the global patterns found in this paper and previous results on the study and modelling of human-flood interactions.

3. Lastly, the first sentence of the Abstract should be improved. "Assessments of potential future flood hazards under climate change using model cascades are subject to large uncertainties, severely limiting our ability to make robust decisions. Instead, here...". I suggest introducing the topic of the study instead of opening with what is not done in this study.

Kind regards,
Giuliano Di Baldassarre

Detailed responses to all comments are provided below. Author response text is in blue and indicated with 'RESPONSE' under each of the reviewer's comments. A tracked changes version of the paper can be found attached with this response. All line and page number references in this response refer to the tracked changes version.

Reviewer 1

The paper addresses the important topic of how (future-) population exposure influences flood risk. This is usually done with a cascade of models, which face a challenge due to uncertainties in parametrizations. The study suggests to use an alternative approach to scenario based studies, and instead apply a stress-testing method under a plausible range of conditions.

The study focuses on stress testing the global river network to quantify and analyse the sensitivity of inundated areas (flood extent) and exposed population to plausible variability in flood magnitude. The analysis reveals regional differences in settlement patterns with respect to flood hazards. It adds to existing studies as to assess the sensitivity of flood hazard and population exposure.

The method used is based on the well-known Fathom model, and an index B showing the inundation extent for all rivers, for each RP: With a power function growth of the flood extent is described with decreasing probability –this is then assumed as an index for sensitivity of flooding. A low value shows areas where flood extent grows rapidly during frequent (low magnitude) events, and vice versa.

Results show that the dominant control on the spatial pattern of floodplain sensitivity is local topography and upstream drainage area. Results show also a relationship between sensitivity and slope steepness – where delta's see the most rapid growth of flood extent during rare events.

The 1000-yr flood extents are then overlaid with worldpop data. The study finds that approximately 2 billion people live on floodplains globally, of which 1.4 billion live on the 100-year floodplain. The study makes a case showing that the type of basins varies regionally (confined, partially confined, etc), and thus the sensitivity also varies regionally. However, much of the population in e.g. Asia lives in unconfined flood plains (delta's) and the research shows these communities are at risk from extreme magnitude floods.

Although it is interesting to see how the authors have classified river reaches, thus far the paper does not show any new insights: steep sections of river have less flood extents and higher frequencies; delta's have larger flood extents related to low probability events.

RESPONSE:

We thank the reviewer for noting the novelty of the population exposure analysis in their subsequent remarks, and we acknowledge that this is likely to be the element of the work with the widest appeal. In addition, however, we believe that there are several important novelties and new insights from the hazard analysis.

Firstly, this is the first global analysis and stress test of a high resolution (~90m) global flood inundation model that includes most of the global river network by considering rivers with an upstream drainage area of 50km² or greater. Other global flood models are much coarser (>1 km) with the minimum catchment size ranging from 500-5000km². Coarser resolution global flood models show substantially higher exposure estimates at all flood magnitudes, but especially for frequent floods, due to people living just off floodplains being included in low resolution inundation extents (Smith et al, 2019). This bias masks the exposure sensitivity to flood magnitude.

Secondly, we use the only global flood model based on a two-dimensional hydrodynamic model for floodplains that is coupled to a one-dimensional model of the river channels. This level of model complexity

is widely regarded as necessary for the accurate simulation of flood events as the river channel is the main conveyor of discharge and interacts with the floodplain in complex ways as flood flows move both from and to the channel (Knight and Shiono, 1996; Fewtrell et al., 2011). Hydrodynamic models without channel-floodplain interactions consistently overpredict inundation from small magnitude floods. The parameterisation of river channel bathymetry must also precisely consider backwater effects that result from non-uniform flows to avoid over-prediction of small floods (See Neal et al., 2021), and although this is widely appreciated in reach scale modelling studies our global flood model is the only one to consider this process. Results from Trigg et al. (2016) indicate that all previous global flood models display very little sensitivity to flood magnitude. These other global flood models simulate floodplain inundation with much simpler methods such as flood volume redistribution (Ward et al., 2013; Winsemius et al., 2013) and floodplain storage elevation relationships (Yamazaki et al., 2011; Pappenberger et al., 2012). The variability in sensitivity observed in our data in relation to floodplain type and region would be absent or substantially weaker when using these older global flood model designs due to the omission of channel-floodplain interaction processes. In the 'Methods Summary' section of the manuscript, we have added a description of the hydrodynamic model and the processes that it is able to simulate (see lines 76-81).

Thirdly, the application of our high resolution, globally consistent modelling framework means that we can robustly show how topography and drainage areas relate to flood sensitivities. We also critically show how flood sensitivities vary for different floodplain types. We believe that this is important given strong regional differences in settlement patterns between the floodplain types.

Finally, it would not be possible to obtain our results from historical observations or other means than modelling. For example, recently published work in Nature has quantified how the population on floodplains has increased over the past two decades using remotely sensed records from MODIS (Tellman et al., 2021). However, this dataset cannot provide information on floodplain sensitivity - as explored here - due to the short record length (18 years), insufficient resolution (250 m), uncertain frequency of missing data in the historical record (due to cloud cover or sub-daily flood maxima going unobserved) and lack of reliable observation capability over vegetated and urbanised areas (Alsdorf et al., 2007).

The interesting part lies when exposure is introduced and the discussion about risk aversion and protection levels. This is done by introducing a parameter that shows how exposed people are distributed across the three flood zones. Figure 3 shows the relation between exposure distribution and flood sensitivity. The 1:1 line represents where population exposure grows at the same rate as the flood hazard with increasing event magnitude.

1. There is maybe a fundamental issue here regarding the method in Fig 3 (which is the central figure of the paper): isn't it too obvious that if a region has a lot of unconfined river reaches, you also may expect a larger proportion of your population living in flood zones that are un-confined? It seems like you have established graph that double counts the flood zone type? So to then simple conclude that those people are risk averse is perhaps too straightforward: maybe they didn't have any choice. As an example: people in the overall unconfined flood zones of Bangladesh can-not all move to India or to the Chittagong hills: are they then risk averse?

RESPONSE:

The reviewer asks a critical question. Our method was specifically designed to overcome the issue of larger proportions of people on larger sections of the floodplain. Several aspects in the design of our analysis prevent this issue from happening, and avoid double counting:

- 1) We separate the analysis of confined, partially confined and laterally unconfined floodplains. The proportion of unconfined floodplains in a region therefore doesn't influence a country's position on

Figure 3, although the total population exposed on the floodplain types governs the size of the markers.

- 2) The relative number of people settled within the frequently or within the rarely flooded zones of a country's floodplain is described by the b_{pop} parameter on the y-axis of Figure 3. Taken on its own, this parameter will be heavily influenced by the floodplain type. For example, a partially confined floodplain will potentially have a very large area flooded during frequent floods relative to rare floods, meaning more people may live within these frequently flooded zones. This leads to a low b_{pop} number (less than 1). By plotting b_{pop} against the flood hazard extent parameter b_r and creating a 1:1 line, we can assess whether the frequently flooded or rarely flooded areas have a higher density of population. Countries above the 1:1 line have higher population densities in areas of the floodplain that flood rarely. Countries below the 1:1 line have higher population densities in areas of the floodplain that flood frequently. Combining both axes on Figure 3 therefore allows us to visualise both the number of people living in areas that flood frequently or rarely, as well as the relative density of these settlements.
- 3) When fitting the power law function to calculate the exposure sensitivity parameters b_{pop} , the population exposed at each return period is normalised to remove the influence of total floodplain population size. Also, when calculating country level mean b_r and b_{pop} values to plot on Figure 3, reaches were weighted equally to avoid a few very populated reaches dominating the analysis.

We strongly believe our approach to the analysis addresses the reviewer's concern. We have made changes to the text (see lines 167-174 in the results section) to significantly simplify the description of the plot, including the interpretation of a country's position on the plot. We can understand why the reviewer found the original version confusing and believe it is now much easier to interpret the plot and understand the data.

Finally, we have also corrected the wording around risk aversion. The reviewer correctly points out that from Figure 3 alone we do not know the reason for a particular settlement pattern. As stated in our overall response, we have improved the discussion of our results using findings from local and regional scale studies. Whether people choose to live in more hazardous areas, or they do so by necessity is a complex question, which we cannot address here. However, we can assess the role that standard of protection (Figure S4) may have played in the settlement patterns and added new plots to the supplementary information (Figures S6 and S7) to extend this discussion to population density.

Other remarks:

2. It took me a while, combining Fig 3 + Fs3, and I believe it shows that people have taken flood protection measures (or spatial planning forced them to live in safer areas with less flood depth?) because they generally are above the 1:1 line -> Can you perhaps address whether this latter statement is correct, and elaborate a bit? Here you could say something about whether spatial planning policies and flood zoning have had prohibiting people from building in dangerous areas?

RESPONSE:

While revising the manuscript we have since noticed that our results section, particularly in the description and explanation of Figure 3, was unnecessarily complex in places. We have now extensively revised this section to simplify these explanations and have now provided a clearer explanation of this key results figure.

In Figure 3 and S4, the 1:1 line represents where population exposure grows at the same rate as the flood hazard, and points close to this line have the same population density in frequently and rarely flooded floodplains. Markers below this line indicate floodplains where the highest density of people is found in the frequently flooded zones of the floodplain. Locations represented by the markers above the 1:1 line indicate floodplains where the highest density of exposed population are found in the rarely flooded areas. We have changed the text in the results section (see lines 180-192) to reflect this explanation.

Another important aspect of Figure 3 is the position of a location on the vertical axis. This location tells us if relatively more people are settled in zones of the floodplain that flood more frequently or rarely. Here we see interesting regional differences for the partially confined floodplains. In Europe and Asia, although the rarely flooded areas are more densely populated (i.e., the position on figure 3 is above the 1:1 line), they have b_{pop} values that are smaller than one, which indicates that in absolute terms more people live in the frequently flooded areas of the floodplain because they cover a greater area. In Europe many studies have shown that flood defence engineering has played a major role in enabling large numbers of people to live in areas of the floodplain that would naturally flood often if undefended. In Asia, both the total population on floodplains and population densities are much greater than in Europe, suggesting that pressure on land has resulted in people living in hazardous areas of the floodplain. We have changed the text in the results section in order to explain these results more clearly (see lines 204-209 for explanation of results in Europe and Asia and see lines 210-215). We also link these results that we find here with several regional and local studies that support our observations (see line 289-304 in the discussion section).

In South America we see contrasting results where more countries have a b_{pop} value greater than 1, which means that more people live on the rarely flooded (less hazardous) areas of the floodplain. We found that although the total floodplain area is large in this region, the population density on floodplains is low which has allowed for preferential settlement in rarely flooded areas of the floodplains (see new Figure S7 in the supplementary information). We have added an explanation for these contrasting results on lines 217-227 in the manuscript, and further discussion on line 305-319.

Unfortunately, data of adequate quality that describe spatial planning policies at the global scale do not yet exist. Even the flood defence database we use is strongly guided by national scale GDP rather than actual flood defence data (Scussoloini et al., 2016). Recent work published in Nature Communications (Rentschler et al., 2022) presents global estimates of the number of people exposed to a 1-in-100-year flood and its interaction with poverty. To do so, a subnational scale GDP and poverty indices dataset is multiplied with exposure headcounts in administrative units to obtain an estimate of the number of people who live below the poverty line and are exposed to flood risk. The authors highlight their assumption that hazard exposure is uniform across income groups within areas, and that an underestimation of flood exposure is likely if there are a disproportionate number of low-income households in riskier, cheaper areas (e.g., if flood risks are reflected in land prices). Currently, we are unable to explicitly explore controls on floodplain settlement patterns with available data at the global scale. It is too coarse to be able to assess processes occurring at fine spatial granularity.

Therefore, we have discussed the settlement patterns we identify in the context of previous local and regional studies where researchers have been able to create suitable datasets. We believe that this is a more robust approach, for now. However, we note in the discussion of the revised manuscript that improving flood defence, spatial planning policy and flood zoning databases would be incredibly useful at reducing the uncertainty in global flood exposure estimates and advancing our understanding of floodplain settlement patterns.

3. In the caption of S3: maybe add that in the left part, the high frequency events dominate, hence the wider 1/10 colours: correct?

RESPONSE:

We agree that this makes the scaling on the plots easier to understand. We have added the following text in the caption: "Note: The scale in (A) differs from that in Figures 3 & S4, where we have used log scales. This is due to the values that b_r and b_{pop} take. When b is less than 1, values range between 0 and 1, however, when b is greater than 1, theoretically values can range between 1 and infinity."

4. Regions above the 1:1 line indicate risk averse populations. I am not sure whether you can make such statement or what you mean by this? Because these location are above the 1:1 line, they have invested more in flood protection, and therefore they are more risk adverse?

RESPONSE:

We agree that our initial use of 'risk averse/risk taking' to describe our results was imprecise, as these conclusions cannot be drawn from Figure 3 alone. Consequently, we have edited the results section to better convey what our results show and what conclusions can be made. See response to comments 1 & 2 for a detailed explanation.

5. Lines 154-157: these are a bit unclear to me, but it also can be the case I have misunderstood the figure 3: "Markers below this line indicate a relatively risk-taking population, where the highest density of the population exposed is in the frequently flooded zones of the floodplain – suggesting that society has taken appropriate steps to protect its population from the flood hazard through levees or other means."  I'd say: the risk takers are indeed the ones who live in the high frequency/low impact zones (this is debatable as well by the way), but then the sentence in addition doesn't fit here ("suggesting society ... other means"). The risk takers usually not invest in flood protection?

RESPONSE:

Since re-reading the results section considering these reviewer comments, we agree that the language used, and the explanations made, were overly complicated. We have therefore removed the text previously shown in lines 154-157. We have also changed our explanation of the relationship between the sensitivity of the physical flood hazard and population densities within the flood hazard zones.

Additionally, in the new lines 174-190, we explain more clearly what Figure 3 describes and what conclusions we can draw from it as discussed above.

6. Lines 167: inclusion of FLOPROS  ...countries that have this patterns..": Which pattern?

RESPONSE:

We have edited this sentence, which now reads: "In Figure S4 (supplement), countries have been sized based on their mean standard of protection extracted from the FLOPROS global flood defence database [46], which reveals that some of these countries that have larger populations on frequently flooded floodplains also have high standards of protection." (lines 200-203).

7. Caption S4: are you suggesting that countries with a higher protection standard are above the 1:1 line?

RESPONSE:

Apologies, the original wording of this section was more complicated than it needed to be. Countries above the 1:1 line have more densely settled populations on the rarely flooded portions of the floodplain. Please see response to questions 2 and 4 for more details.

8. Line 363: did you run a sensitivity analysis for this power-function: I'd like to see how this plays out for the position of the 1:1 line?

RESPONSE:

We have not run an explicit sensitivity analysis for the power law function that we use to quantify the sensitivity of flooded area and population exposure to return period. However, we do not believe it is necessary for the following reasons:

- 1) By normalising flood extents, population totals and return periods, we can fit a power law function where its exponent is the only variable. This power law has to fit a small number of points on each reach (10 as this is the number of flood hazard maps that are produced by the global flood model). We used a very strict goodness-of-fit threshold to decide whether the power law is a good descriptor of the shape the data produce or not. The power law fit must have an R^2 value greater than 0.9. Fitting a one-parameter function to a small dataset with very distinct shapes does not leave space for relevant error, which our visual assessment for selected examples confirmed. The very high R^2 threshold ensures that we eliminate any cases where the power law would not be a good fit (and the b parameter would thus not be meaningful or robust).
- 2) We average over many reaches (an average of 835, 4800 and 272 for confined, partially confined and laterally confined floodplains, respectively per country) for much of our analysis so we expect that any remaining variability for individual power law fits would average out when producing figure 3. We believe our country level results are robust due to the volume of data aggregated.

9. I am not sure what the authors mean by the sentence in lines 43-45. I expect a response to the previous paragraph, where the cascading modelling approach is challenged due to modelling biases. But then lines 43-45 speak about the need to “...identify exposure of populations and assets across a spectrum of plausible events to derive actionable information under large uncertainty”.  This is also possible with a modelling approach. I'd even argue that with a scenario approach, you may assess an even larger range of uncertainty.

RESPONSE:

We agree that this sentence was not clear and have removed it to improve the storyline of our introduction.

For clarity, various researchers have proposed a strategy to deal with poorly determined uncertainties (so-called deep uncertainties) as we regularly find in the context of climate change impact studies. These approaches have been referred to as scenario discovery or bottom-up approaches – in contrast to top-down scenario-based modelling (Brown et al., 2012, WRR). The underlying idea is to sample the feasible input space, rather than the input space defined by one or more scenarios. By definition, this approach should therefore include more uncertainty than a scenario-based strategy (or at least not less). This approach is widely accepted as an alternative strategy to account for input uncertainty and to provide helpful information despite significant uncertainty (e.g. Quinn et al., 2020, Earth's Future). Our approach uses an existing large and coherent global set of flood estimates up to 1,000-year return period as basis for such a bottom-up strategy, here implemented in the form of a stress-test.

10. Lines 124-127: “Populations on partially confined floodplains are most likely to experience vulnerability to increasing intensity and magnitude of frequent flood events, whereas confined and laterally unconfined floodplains are more vulnerable to the increasing probability of extreme flood events”

 I don't see the link directly? Doesn't the vulnerability it also depend on how much flood protection has been installed?

 I would make a difference between sensitivity and vulnerability: these are two different concepts.

RESPONSE:

We agree that 'vulnerability' was not the correct use of the word here, particularly when discussing flood hazard, risk, and exposure. To improve the explanation of the key point conveyed here, we have removed this sentence. Instead, we have changed the text to now read as (lines 143-149): "Populations on partially confined floodplains are most likely to be susceptible to increasing intensity of frequent flood events. and They might therefore be impacted by deeper flood waters, such as those occurring in cases of overtopping of defences, leading to significant potential for damages and economic losses. While populations on laterally unconfined and confined reaches are more likely to be impacted by changes in the probability of extreme rare floods which would reach parts of the floodplain which were likely not inundated in living memory. Thus, providing the potential for catastrophic and unanticipated impacts."

11. Why didn't you include flood depth? Now you see in Figure 2 that a large share of the people in SE Libya live on flood plains: but are these areas in the middle of the Sahara really hazardous flood plains?

RESPONSE:

This comment asks two questions, and so we will respond to each separately.

- 1) Our decision to use flood extents and to not include flood depth is due to the vertical inaccuracy of the DEM used (Hawker et al., 2018). Flood depths produced by global flood models are less reliable at the pixel scale. To use flood depths within our methodology, i.e., to assess the sensitivity of floodplains to varying event magnitudes, we would have to produce a meaningful reach-scale metric of flood depth for each of the return periods. This is difficult due to the locally varying depths that are driven by topographic variability. And, when averaging depths over a large area, these may look small due to the small values of the fringe flooding that dominates.
- 2) Dryland rivers are often ephemeral, however that doesn't mean they do not flood. Furthermore, although population numbers in dryland basins are often low, we see a high proportion of the population living on floodplains. We therefore believe that it's important to include these areas in our results. Nevertheless, analysis of the extreme flows used in our model, and several other flood models, have shown that the uncertainty in extreme river discharge is greater in arid catchments (Devitt et al., 2021). Validation work on our modelling system over the US has also demonstrated that the accuracy of the flood hazard model is lowest in arid areas relative to other climate zones (Wing et al., 2017). We have added this issue to our 'implications for future research and key methodological uncertainties' section (lines 406-409).

Reviewer 2

The paper by Devitt et al. analysis global patterns of human settlements in flood-prone areas. They found population to be more densely settled in areas prone to rare (but potentially catastrophic) floods, and more evenly distributed exposure in areas exposed to frequent (mostly low magnitude) floods. This work is technically sound and the paper is nicely written. I found these results not only scientifically relevant, but also socially important. They have the potential to substantially contribute to advance our fundamental knowledge on flood risk dynamics. Here, I provide three main suggestions to further enrich the description of this global study.

RESPONSE:

We thank reviewer 2 for their positive assessment of the manuscript. Please see our detailed responses to comments below.

1. The outcomes of this paper are strongly related to recent findings on human-flood interactions. For example, the AGU journal Water Resources Research published "Debates –Perspective on Socio-Hydrology" in 2015 to stimulate discussion on frontier research. These debates include an original research article, four

commentaries, and an editorial. The former provides case studies and model results of two dynamics, i.e. adaptation and levee effect, which strongly relates to the main outcomes of this paper. I think this paper would highly benefit from including a paragraph in the introductory part describing these general dynamics of human-water systems.

RESPONSE:

Thank you for directing us to these studies. It is interesting to see how dynamics that are observed in location specific case studies can also be seen in the outcomes of our work at the global scale. We added detail to the introduction of how humans have been co-evolving with flood hazards for a long time and that many researchers have used country-specific and regional scale case studies to understand the emergent dynamics (see line 65-74). We have added details from these case studies to our discussion to connect location specific and global analyses. We believe this inclusion has significantly strengthened our argument and discussion. See also our response to the next reviewer comment.

2. To give perspective to these outcomes, I think it important to discuss their implications for the ongoing development of theories of human-flood interactions. As stated in the previous point, these global patterns nicely complement numerous case studies and system dynamic models carried out by the several scholars, such as the ones engaged with the international initiative Panta-Rhei-Everything Flows (IAHS). I suggest to add one paragraph and discuss more explicitly similarities and differences between the global patterns found in this paper and previous results on the study and modelling of human-flood interactions.

RESPONSE:

We have extended the discussion section by adding direct comparisons with other studies that have looked at floodplain settlement patterns. In the discussion section (lines 289-304) we have compared our results presented for countries in Asia with local studies that have assessed the preferential development of floodplains in this region. We find that the high exposure that we find in the high hazard zones of the floodplains has been found in studies that have looked at how urban expansion in this region has led to safe locations becoming increasingly scarce, and this has caused development of high-risk areas in order to match the high pressure of rapid population growth. In the discussion section (lines 305-319) we have compared our findings for South America with case studies that have found that the regular experience of flood rich periods has allowed for relatively hazard averse settlement patterns. We have linked this with the “adaptation effect” and a case study example that observed this dynamic in the Paraná River basin. We have also added to the ‘implications for future research and key methodological uncertainties’ section (lines 385-389) which looks at two typical examples – Bangladesh and Vietnam - with a long history of living and evolving with flooding. People in these places have adapted to the threat of regular flooding by using it to their advantage by adjusting economic activities through agriculture and fisheries.

We believe that these additions have strengthened our discussion of our results and by comparing them more directly to other studies has highlighted the complimentary nature of our findings with existing understanding of human-water dynamics obtained from detailed location specific analysis.

3. Lastly, the first sentence of the Abstract should be improved. “Assessments of potential future flood hazards under climate change using model cascades are subject to large uncertainties, severely limiting our ability to make robust decisions. Instead, here...”. I suggest introducing the topic of the study instead of opening with what is not done in this study.

RESPONSE:

We agree. We have changed the opening sentence of the abstract to give a statement of the problem and the way that we aim to address this in our work. This now reads: “Flooding is one of the most common natural hazards, causing disastrous impacts worldwide. Stress-testing the global human-Earth system to understand the sensitivity of floodplains and population exposure to a range of plausible conditions is one strategy to identify where future changes to flooding or exposure might be most critical.”

References:

- Alsdorf, D., Rodríguez, E., & Lettenmaier, D. (2007) Measuring surface water from space, *Reviews of Geophysics*, 45(2), 2006RG000197
- Bates, P. D., Quinn, N., Sampson, C., Smith, A., Wing, O., Sosa, J., et al. (2021). Combined modeling of US fluvial, pluvial, and coastal flood hazard under current and future climates, *Water Resources Research*, 57(2), e2020WR028673
- Brown, C., Ghile, Y., Laverty, M. & Li, K. (2012) Decision scaling: Linking bottom-up vulnerability analysis with climate projections in the water sector, *Water Resources Research*, 48, W09537
- Devitt, L., Neal, J., Wagener, T. & Coxon, G. (2021) Uncertainty in the extreme flood magnitude estimates of large-scale flood hazard models, *Environmental Research Letters*, 16, 064013
- Fewtrell, T.J., Neal, J.C., Bates, P.D., & Harrison, P.J. (2011) Geometric and structural river channel complexity and the prediction of urban inundation, *Hydrological Processes*, 25(20), 3173-3186
- Hawker, L., Rougier, J., Neal, J., Bates, P., Archer, L. & Yamazaki, D. (2018) Implications of simulating global digital elevation models for flood inundation studies, *Water Resources Research*, 54(10), 124032
- Knight, D.W. & Shiono, KL (1996) River channel and floodplain hydraulics. In W. Anderson, & Bates (Eds.), *Floodplain processes* (pp. 139-181). New York, NY: Wiley
- Neal, J., Hawker, L., Savage, J., Durand, M., Bates, P., & Sampson, C. (2021). Estimating river channel bathymetry in large scale flood inundation models. *Water Resources Research*, 57, e2020, WR028301
- Pappenberger, F., Dutra, E., Wetterhall, F., & Cloke, H.L. (2012) Deriving global flood hazard maps of fluvial floods through a physical model cascade, *Hydrology and Earth System Sciences*, 16, 4143-4156
- Quinn, J.D., Hadjimichael, A., Reed, P.M. & Steinschneider, S. (2020) Can exploratory modeling of water scarcity vulnerabilities and robustness be scenario neutral? *Earth's Future*, 8(11), e2020EF001650
- Rentschler, J., Salhab, M. & Jafino, B.A. (2022) Flood exposure and poverty in 188 countries, *Nature Communications*, 13, 3527
- Scussolini, P., Aerts, J.C.J.H., Jongman, B., Bouwer, L.M., Winsemius, H.C., de Moel, H. & Ward, P.J. (2016) FLOPROS: an evolving global database of flood protection standards, *Natural Hazards and Earth System Sciences*, 16, 1049-1061
- Smith, A., Bates, P.D., Wing, O., Sampson, C., Quinn, N. & Neal, J. (2019) New estimates of flood exposure in developing countries using high-resolution population data, *Nat Communications*, 10, 1814
- Tellman, B., Sullivan, J.A., Kuhn, C., Kettner, A.J., Doyle, C.S., Brakenridge, G.R., Erickson, T.A., & Slayback, D.A. (2021) Satellite imaging reveals increased proportion of population exposed to floods, *Nature*, 596, 80-86
- Trigg, M.A. et al. (2016) The credibility challenge for global fluvial flood risk analysis, *Environmental Research Letters*, 11(9), 094014

Ward, P.J., Jongman, B., Weiland, F., Bouwman, A., van Beek, R., Bierkens, M., Ligtoet, W. & Winsemius, H.C. (2013) Assessing flood risk at the global scale: model setup, results, and sensitivity, *Environmental Research Letters*, 8, 044019

Wing, O. E. J., Bates, P. D., Sampson, C. C., Smith, A. M., Johnson, K. A., & Erickson, T. A. (2017), Validation of a 30 m resolution flood hazard model of the conterminous United States, *Water Resources Research*, 53, 7968– 7986

Winsemius, H.C., van Beek, L.P.H., Jongman, B., Ward, P., & Bouwman, A. (2013) A framework for global river flood risk assessments, *Hydrology and Earth System Sciences*, 17, 1871-1892

Yamazaki, D., Kanae, S., Kim, H., & Oki, T. (2011) A physically based description of floodplain inundation dynamics in a global river routing model, *Water Resources Research*, 47, W04501

REVIEWERS' COMMENTS

Reviewer #1 (Remarks to the Author):

I have carefully read the revisions of the paper "Flood hazard potential reveals global floodplain settlement patterns"

In particular, the explanation of Fig. 3 has been improved, and other issues around the use of risk aversion and vulnerability have been addressed.

I think the revisions are convincing and the paper now makes a strong contribution for the readership of Nature Com.

I recommend publication

Reviewer #2 (Remarks to the Author):

My concerns have been addressed in the revision process.

REVIEWERS' COMMENTS

Reviewer #1 (Remarks to the Author):

I have carefully read the revisions of the paper "Flood hazard potential reveals global floodplain settlement patterns"

In particular, the explanation of Fig. 3 has been improved, and other issues around the use of risk aversion and vulnerability have been addressed.

I think the revisions are convincing and the paper now makes a strong contribution for the readership of Nature Com.

I recommend publication

We thank the reviewer for their time given to the peer review of our work.

Reviewer #2 (Remarks to the Author):

My concerns have been addressed in the revision process.

We thank the reviewer for their time given to the peer review of our work.